# Transcription decouples estrogen-dependent changes in enhancer-promoter contact frequencies and spatial proximity

**Luciana I. Gómez Acuña** [ID], **Ilya Flyamer**[¤], **Shelagh Boyle, Elias T. Friman** [ID], **Wendy A. Bickmore** [ID]*

MRC Human Genetics Unit, Institute of Genetics and Cancer, University of Edinburgh, Crewe Road, Edinburgh, United Kingdom

¤ Current address: Friedrich Miescher Institute for Biomedical Research, Basel, Switzerland
* Wendy.Bickmore@ed.ac.uk

**Data Availability Statement:** Data and statistical analysis supporting this article are available in the accompanying Supplementary Data. TT-seq and C-TALE data have been deposited in NCBI GEO under

## Abstract

How enhancers regulate their target genes in the context of 3D chromatin organization is extensively studied and models which do not require direct enhancer-promoter contact have recently emerged. Here, we use the activation of estrogen receptor-dependent enhancers in a breast cancer cell line to study enhancer-promoter communication at two loci. This allows high temporal resolution tracking of molecular events from hormone stimulation to efficient gene activation. We examine how both enhancer-promoter spatial proximity assayed by DNA fluorescence in situ hybridization, and contact frequencies resulting from chromatin in situ fragmentation and proximity ligation, change dynamically during enhancer-driven gene activation. These orthogonal methods produce seemingly paradoxical results: upon enhancer activation enhancer-promoter contact frequencies increase while spatial proximity decreases. We explore this apparent discrepancy using different estrogen receptor ligands and transcription inhibitors. Our data demonstrate that enhancer-promoter contact frequencies are transcription independent whereas altered enhancer-promoter proximity depends on transcription. Our results emphasize that the relationship between contact frequencies and physical distance in the nucleus, especially over short genomic distances, is not always a simple one.

## Author summary

Investigating the three-dimensional organization of the genome is important for understanding how enhancers regulate their target genes. Commonly, exploring 3D genome organization uses either an imaging-based method—DNA fluorescence in situ hybridization, or molecular chromosome conformation capture methods. Here we use a cell system of nuclear hormone induced enhancer and gene activation to compare what insight into enhancer–gene promoter interaction these methodologies provide. To our surprise, we found that these two methods can produce paradoxical results, with increased enhancer-promoter contact frequencies assayed by chromosome conformation capture occurring in

accession numbers GSE225508 and GSE225617 respectively

**Funding:** L.I,GA, S.B, W.A.B. are supported by MRC University Unit grants MC_UU_00007/2 and MC_UU_00035/7. E.T.F:Swiss National Science Foundation (P500PB_206805).

**Competing interests:** The authors have declared that no competing interests exist.

a situation where imaging shows overall decreased enhancer-promoter proximity. We explore what might contribute to these seemingly discrepant data, demonstrating that whereas increased enhancer-promoter contact frequencies require binding of an active transcription factor to the enhancer but are independent of transcription per se, altered enhancer-promoter proximity depends on transcription. Our analyses show the importance of using orthogonal methods to fully explore 3D genome organization.

## Introduction

How enhancers exert their action over long genomic distances is not understood. Contemporary models range from those that involve direct enhancer-promoter contact in 3D space, models that do not require enhancer-promoter contact with each other but do require engagement with transcriptional hubs, through to models based on diffusion of activating signals in a confined volume of the nucleus [1,2].

Two major methodologies–biochemical and imaging—are most often used to study the proximity of enhancers in relation to the promoters they regulate. Chromosome conformation capture (3C)-derived techniques rely on enzymatic fragmentation of cross-linked and detergent-treated chromatin followed by proximity ligation. When followed by next-generation sequencing, techniques such as Hi-C and micro-C can provide pair-wise contact frequencies genome-wide. Hundreds of thousands, to millions, of cells are often required to obtain high-resolution interaction maps, so the resulting data represent average contact frequencies of the cell population.

In imaging-based methods, genomic loci are visualized either in live or fixed cells. Techniques such as DNA fluorescence in situ hybridization (DNA FISH) generally allow for analysis of only a handful of loci at a time in a few hundred cells but distances between loci in the nucleus can be determined at high spatial resolution at the single cell/allele level.

Generally, 3C and imaging-based approaches give concordant views of 3D genome organization [3–5]. However, there are examples where imaging data do not match the expectations from proximity ligation frequencies [3,6], suggesting that the mechanistic basis for both data sets warrants further examination [7].

A particularly fascinating area of dynamic 3D genome organization is enhancer-promoter interaction. Hi-C and Micro-C data both show evidence for enriched interactions between enhancers and their target promoters in vertebrate genomes [8,9]. These data are consistent with a looping model of enhancer-promoter direct contact [2]. In some cases an increased frequency of enhancer-promoter colocalization has been reported by imaging, consistent with enhancer-promoter looping [10,11]. However, a number of other observations from imaging techniques are not consistent with a simple enhancer-promoter contact model where enhancers are juxtaposed to, or contact, their cognate target genes in a somewhat stable manner. Both live-cell imaging and DNA FISH in fixed cells provide no evidence that very close proximity (<200nm) between enhancers and promoters is temporally correlated to enhancer-driven gene transcription in mouse embryonic stem cells (mESCs) [12–14]. Indeed, DNA FISH indicates reduced enhancer-promoter proximity upon enhancer activation at the *Shh* locus [13,15].

Enhancers in the complex regulatory landscapes of developmental genes may have different mechanisms of action from those that function in other biological contexts, such as in the acute responses to physiological cues. Estrogen-receptor alpha (ERα)-dependent enhancers in the human breast cancer cell line MCF-7 allow tracking of events after hormone treatment at

high temporal resolution in a well-studied model system in which enhancers and their target genes, as well as many of the molecular mechanisms operating during enhancer activation, are well defined [16]. Chromosome conformation capture assays including 3C, Hi-C and ERα-selected CHIA-PET have been used to examine contact frequencies in MCF-7 cells before and after the addition of the ERα ligand 17β-estradiol (E2) [17–19]. By contrast, there has been little visual examination of the spatial proximity of specific ERα-enhancers and their target genes in the nucleus.

Here we use Chromatin TArget Ligation Enrichment (C-TALE) and DNA FISH in MCF-7 cells to examine the high-resolution 3D organization of two genes whose transcription is induced rapidly and efficiently from ERα-bound enhancers upon E2 addition. We demonstrate that the two orthogonal methods produce seemingly paradoxical views of 3D enhancer-promoter organization in response to induction and we explore the basis for these apparent discrepancies using different ligands of ERα and inhibitors of transcription. Our data demonstrate the need to carefully explore the role of enhancer-bound complexes, RNA polymerase II and transcription as determinants of 3C contact frequencies and the enhancer-promoter spatial distances determined by DNA FISH.

## Results

### Rapid enhancer-dependent gene activation in response to estradiol

ERα is a member of the nuclear receptor transcription factor family; upon binding to the ligand E2 it is recruited to specific pre-marked regulatory regions (including enhancers) within minutes [20], inducing rapid transcription of target genes [21]. Taking advantage of the extensive data available on transcription factor and co-activator chromatin binding in MCF-7 cells stimulated with E2, we identified two enhancer-gene pairs to study enhancer-promoter communication. *GREB1* and *NRIP1* are well studied E2 responsive genes [21]. E2 responsive enhancers were identified using published ERα and p300 binding data from untreated and E2 treated cells [16,18,22], and were paired to the target genes by proximity. A putative *GREB1* enhancer is located 40 kb upstream of the gene promoter and the *NRIP1* enhancer is located 100 kb upstream of the promoter. Both of these sites acquire chromatin accessibility, as assayed by ATAC-seq, upon E2 treatment [23] (Fig 1A).

Using nascent RNA-seq (TT-seq) [24], 5, 30 and 60 min after E2 addition to MCF-7 cells which had been extensively starved of hormone, we detect abundant transcription from these genes 30 mins after E2 addition, but not at the earlier 5-minute time point (Fig 1A, 1B and S1 Table). As previously reported using GRO-seq [18], TT-seq also reveals enhancer RNAs (eRNAs) transcribed at these ERα-bound sites and induced with similar kinetics to the gene mRNAs (S1A and S1B Fig and S1 Table). There seems to be a unidirectional long transcript running through the intergenic space upstream of *GREB1*, for *NRIP1* short eRNAs are detected at the sites of ERα-binding.

We confirmed these transcription dynamics at the level of individual alleles by nascent RNA FISH, using probes targeted to the first intron of each gene. Pre-mRNA FISH detects significant upregulation of *GREB1* 10 min after hormone addition, with the frequency of active alleles increasing up to an hour after E2 addition when 79% of alleles are found to be active (Figs 1C; S1C and S1 Table). There are 4 copies of *GREB1* in our MCF-7 cells—two copies on cytologically normal chromosomes 2, one on a chromosome with additional material translocated onto 2q and the fourth on a small portion of chromosome 2p translocated onto another chromosome [25]. The high proportion of cells with 3 or 4 foci of *GREB1* RNA FISH signal 20–40 mins after E2 addition indicates that all 4 GREB1 alleles are induced by E2. *NRIP1* is on chromosome 21 of which there are 2 copies in MCF-7 cells. Significantly increased *NRIP1*

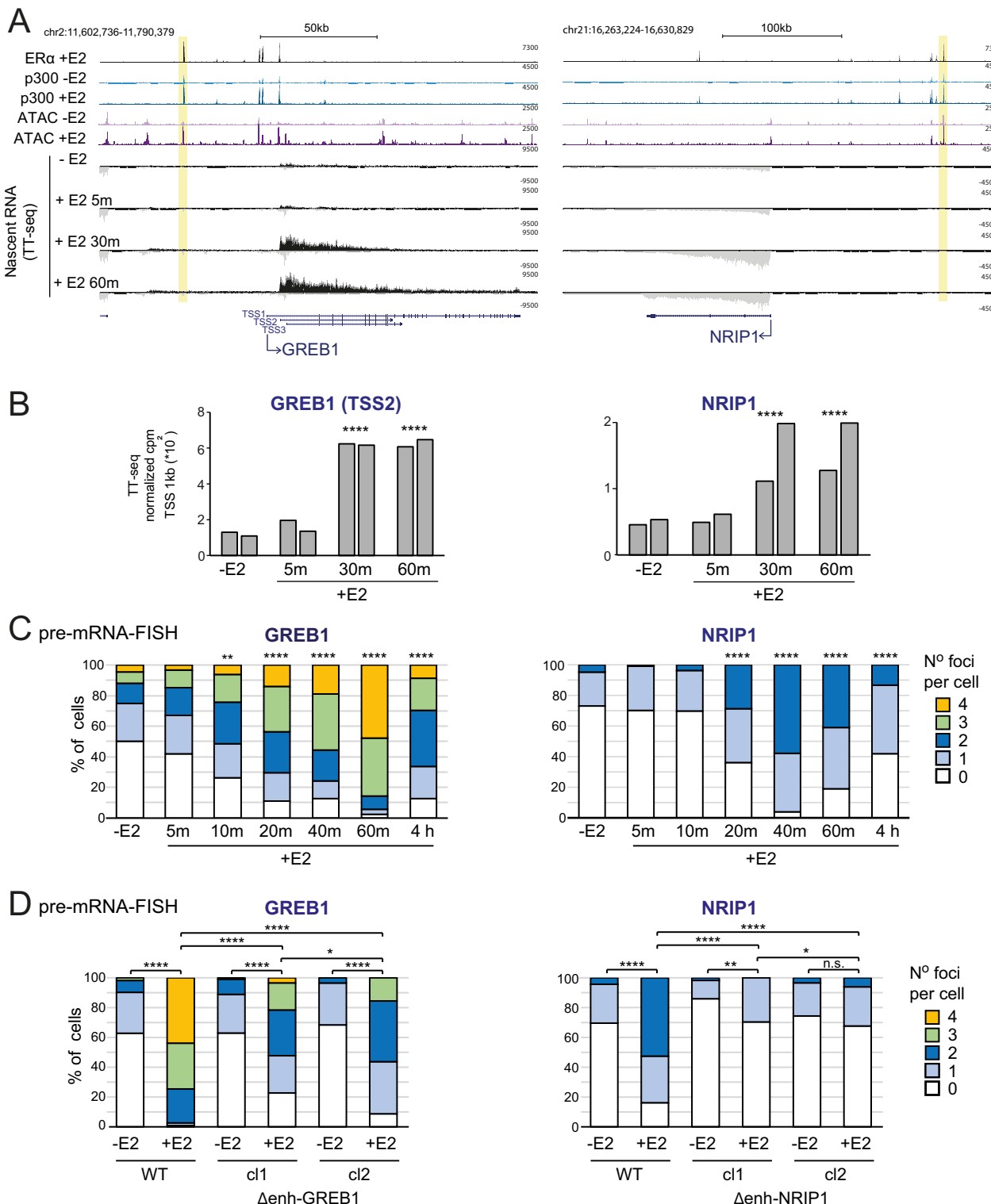

**Fig 1. Rapid induction of ER-responsive genes.** A) Genome browser snapshots of the *GREB1* (left) and *NRIP1* (right) loci showing published ChIP-seq tracks of ERα in MCF-7 cells after E2 addition, p300 with and without E2 [16], ATAC-seq with and without E2 [23] and TT-seq tracks without E2 and 5, 30 and 60 min after E2 addition. Yellow bars indicate putative ERα enhancers. Genome coordinates: hg19 assembly of the human genome. B) Quantification of TT-seq reads over 1kb regions extending downstream of *GREB1* and *NRIP1* TSSs. Normalized counts per million reads (cpm) of two replicates are shown. C and D) pre-mRNA FISH for *GREB1* and *NRIP1* nascent transcripts without and with E2 in (C) MCF-7 cells for the indicated

timepoints or (D) for the 1h timepoint in cells where the respective putative ERα enhancers have been deleted. Results for two independent homozygous clones are shown. The percentage of cells with 0, 1, 2, 3, 4 foci is shown. Two-sided Fisher exact test. *p<0.05, **p<0.01, ***p<0.001, ****p<0.0001. Biological replicates for the data in panels C and D are in S1 Fig. Statistical data for Fig 1 are in S1 Table.

transcriptional activation is observed after 20 min of stimulation, with levels reaching a maximum at 40 mins when 73% of alleles appear to be active (Figs 1C; S1C and S1 Table). For both genes, the number of active alleles decreases by 4h after stimulation, but not down to the level of untreated cells.

The putative *GREB1* and *NRIP1* ERα enhancers have other features, such as p300 and FOXA1 occupancy [18,26], expected of a functional enhancer, but direct genetic evidence linking these sites to the activity of target genes has been lacking. To address this, we used CRISPR-Cas9 with guides targeting the borders of the ERα peaks to delete these putative enhancers in MCF-7 cells. Two clones bearing homozygous deletions for each of the two enhancer regions were recovered (S2A Fig). GREB1 enhancer deletion led to a significant decrease in the efficiency of *GREB1* activation in E2-treated cells detected by nascent RNA FISH (Figs 1D; S2B and S1 Table). Residual *GREB1* expression may result from multiple regulatory regions coordinating *GREB1* expression in E2 treated cells [27]. Deletion of the NRIP1 enhancer led to a dramatic decrease of *NRIP1* induction in response to E2 (Figs 1D; S2B and S1 Table). In contrast, *NRIP1* was still highly E2 inducible in cells deleted for the GREB1 enhancer, and *GREB1* E2-dependent induction was unaffected by deletion of the NRIP1 enhancer (S2C Fig and S1 Table). These data confirm the role of the upstream ERα binding sites as *bona fide* enhancers driving E2-responsive transcription at *GREB1* and *NRIP1*.

## Estrogen induces increased enhancer-promoter contact frequency, but decreased proximity

Whether enhancers drive expression of their target promoters through a direct physical interaction, or through contact-independent mechanisms, is often assayed from the interaction frequencies derived from chromosome conformation capture assays [28]. To investigate E2-dependent enhancer-promoter interaction, we performed high resolution Hi-C analysis by Chromatin TArget Ligation Enrichment (C-TALE) [29], using tiled BACs to enrich the 500 kb region at *GREB1* and a 1 Mb region at *NRIP1*, in MCF-7 cells treated with vehicle, or with E2 at 5, 30 and 60 minutes. Relative enhancer-promoter contact frequencies increase upon E2 stimulation at both loci, with enhanced focal contacts detectable between the enhancer and promoter regions (Figs 2A and S3A, arrowheads). Using the enhancer or promoter regions as viewpoints in virtual 4C plots, increased enhancer-promoter contact frequencies can be detected as early as 5 min after E2 stimulation (Figs 2B and S3B, arrowheads) and are the highest in the whole captured regions (S3C Fig). Most noticeable in the case of *GREB1*, transcription-related changes along the length of the gene body are also observed: while very short-range relative contact frequencies increase along the gene when it is transcribed, longer-range contact frequencies decrease (arrowed in Figs 2A and S3A). We note that the C-TALE data indicates that there are no large genomic rearrangements within the *GREB1* locus on any of the 4 alleles in MCF-7 cells.

Imaging-based methods, such as DNA FISH and live cell imaging, have questioned whether enhancer-promoter juxtaposition is required for transcription [12,13,15], including for enhancer activation by nuclear hormone-dependent enhancers [30]. To explore this in the context of rapid gene induction from ERα-dependent enhancers, we used DNA FISH with fosmid-derived probes spanning the promoter, enhancer and control regions at the *GREB1* and *NRIP1* loci (S3C Fig). In the case of *GREB1*, the DNA FISH probes were chosen to maximize

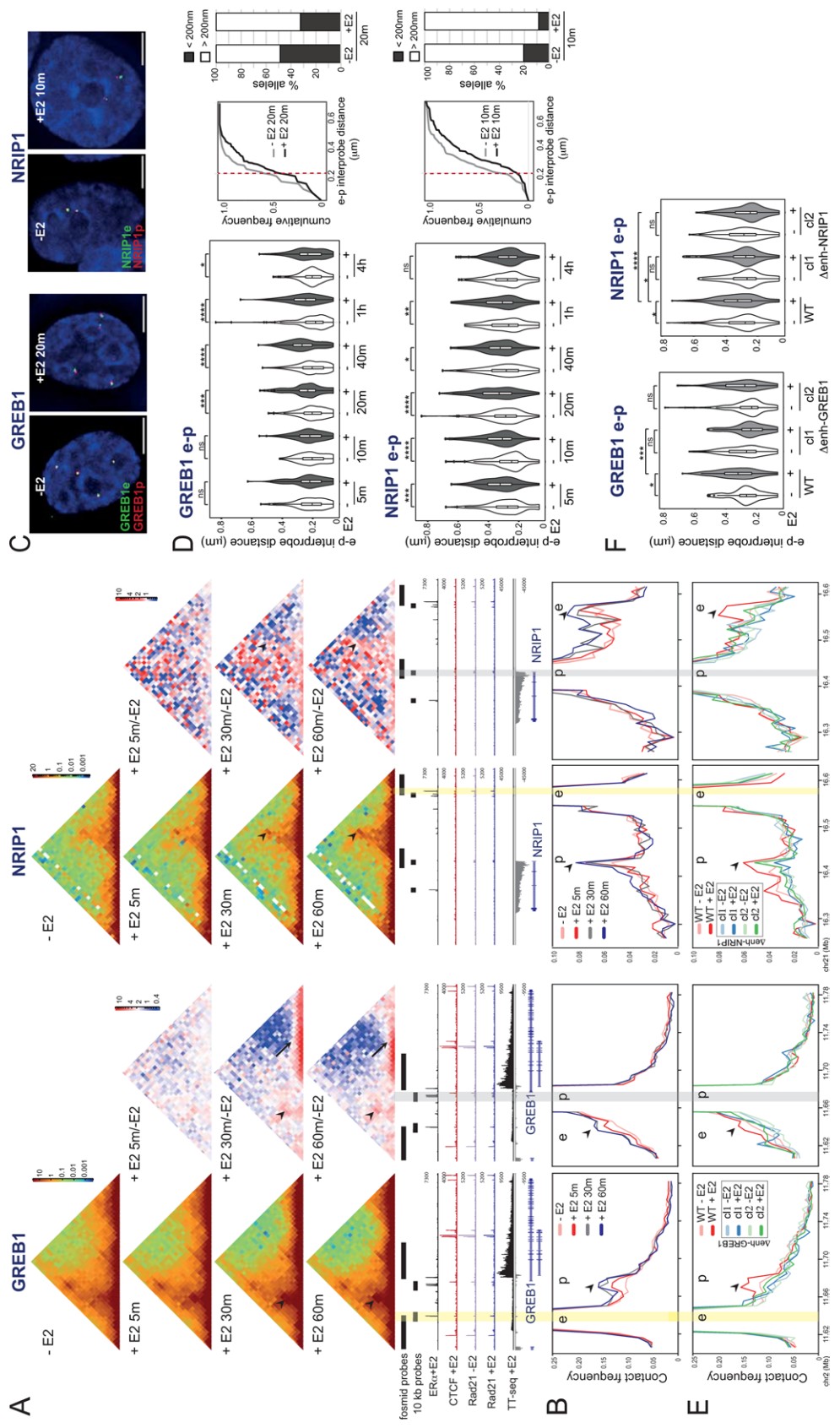

**Fig 2. E2 induces increased chromosome conformation capture contact frequencies but decreased proximity between enhancers and promoters.** A) C-TALE heatmaps at *GREB1* (left; 5kb resolution) and *NRIP1* (right; 10kb resolution) loci from MCF-7 cells without (-E2) and with E2 (+E2) for the indicated time points. ERα +E2 [16], CTCF +E2, Rad21 −E2, Rad21 +E2 ChIP-seq tracks [49], and TT-seq +E2 30 min are shown below. Enhancer regions are highlighted with yellow bars. In red-green-blue heatmaps (left), each pixel represents the normalized contact frequency between a pair of loci. In red-white-blue heatmaps (right) each pixel represents the ratio between E2-treated at the indicated time points and untreated samples. Blue indicates loss of contact frequency in treated vs untreated samples and red pixels indicate a gain. Arrowheads indicate pixels corresponding to enhancer-promoter pairs. Arrows indicate the enriched short-range contacts along the transcribed *GREB1* gene body. Data from an independent biological replicate are in S3A Fig. B) Virtual 4C plots derived from the normalized contact frequencies in (A) using the enhancer (left) and promoter (right) regions as viewpoints (yellow and grey bars, respectively). Genome coordinates: hg19 human genome. Arrowheads indicate the (left) promoters (p), or (right) the enhancers (e) of *GREB1* and *NRIP1*. C) Representative images of nuclei (DAPI, blue) from untreated (-E2) or MCF-7 cells treated for the indicated times with E2 showing DNA FISH signal from fosmid probes targeted to the enhancer (e, green) and promoter (p, red) regions of the *GREB1* (left) or *NRIP1* (right) loci. The position of the probes is shown in (A) and in S3C Fig. Scale bars, 5μm. D) Left: Violin plots showing the distribution of DNA FISH inter-probe distances (μm) between e-p fosmid probe pairs at *GREB1* (top) or *NRIP1* (bottom) in untreated and E2 treated MCF-7 cells for the indicated time points. Boxes indicate median and interquartile distances. The statistical difference in data distribution +/-E2 at each time point was assessed by a two-sided Mann-Whitney test. n.s. p>0.05 *p<0.05, **p<0.01, ***p<0.001, ****p<0.0001. Holm-Bonferroni correction for multiple testing. Data from an independent biological replicate are in S3D Fig. Middle: Cumulative frequency plots of e-p inter-probe distances for the indicated time points and loci. Red dashed line indicates 200nm Right: proportion of e-p inter-probe distances for the indicated timepoints that are < or > that 200 nm. E) As in (B) but for WT and enhancer deletion clones. Heatmaps and data from an independent biological replicate are in S5 Fig. F) As in (D) but for WT and enhancer deletion clones untreated and E2 treated for 30 min. Data from an independent biological replicate are in S5 Fig. Statistical data for Fig 2 are in S2 Table.

their genomic distance to aid spatial resolution. We measured inter-probe distances at different time points after E2 stimulation. At both loci, enhancer-promoter inter-probe distances increase upon E2 treatment. In the case of *GREB1* this is statistically significant 20 min after stimulation, but the trend is already observable at 10 min (Figs 2C, 2D, S3D and S2 Table). At *NRIP1*, significantly increased enhancer-promoter separation is observed as soon as 5 min after hormone stimulation, is sustained until 1 hour after E2 addition, but is no longer observed at 4h. Distances between enhancers and control probe pairs do not change in response to E2 (S4A and S4B Fig and S2 Table).

It is usually thought that regions showing high contact frequencies in 3C techniques fall within a 200 nm radius of each other and that, rather than looking at overall distance distributions and median values in imaging data, the proportion of loci at short distances is a more appropriate comparison [7,31]. However, despite the increased enhancer-promoter contact frequencies we detect by C-TALE after E2 addition, cumulative frequency analysis at both the *GREB1* and *NRIP1* regions show that the proportion of alleles with enhancer-promoter inter-probe distances <200 nm decreases in E2 treated cells (Fig 2D, middle and right panels panels). No difference is observed for enhancer-control probe cumulative distributions upon E2 addition (S4A and S4B Fig).

To ensure that any very focal enhancer-promoter co-localisation was not being obscured by our use of comparatively large (40kb) fosmid-derived probes, we repeated DNA FISH with probes detecting 10 kb regions centred on the sites with E2-triggered C-TALE higher contact frequencies (S4C and S4D Fig). It should be noted that, in the case of NRIP1, the enhancer region involved in the highest contact frequency does not fall precisely on the main ERα peak, but over the nearby smaller peaks—visible in the virtual 4C plots with the *NRIP1* promoter as the viewpoint (Figs 2B and S3B, right panels). We also included a probe targeted to an *NRIP1* intragenic (i) ERα ChIP peak (S3C Fig). With these small probes, we still observe that enhancer-promoter distances increase, and the proportion of alleles with enhancer-promoter distances <200nm decreases, upon E2 addition at both *GREB1* and *NRIP1* loci (S4C and S4D Fig and S2 Table). No change in distance is seen between the enhancers and control probes.

The absolute distances measured with the 10kb probes are smaller than those measured using the larger fosmid probes. This may be partly due to to use of a super-resolution imaging platform to image the signals from the smaller probes, providing improved resolution, especially in the *z* dimension, compared to the faster epifluorescence imaging platform used for all other experiments. With this improved resolution, we note that that the high proportion of alleles with enhancer-promoter distances <200nm in the absence of E2 (S4C and S4D Fig) may indicate a poised conformation.

*GREB1* and *NRIP1* are strongly activated by E2, with only very low level transcription detected in the absence of E2 (Fig 1). To ascertain if decreased enhancer-promoter proximity is also seen at other E2 responsive genes, we examined the *CCND1* locus. *CCND1* is transcribed convergently toward the widely expressed neighbouring *LTO1* whose expression is not affected by E2 (S5A and S5B Fig). *CCND1* encodes Cyclin D1, which drives the G1/S transition in many cell types. *CCND1* expression is detected in the absence of E2, but is further upregulated 30 mins after of E2 addition (S5A and S5B Fig and S2 Table) at which time transcription can also be detected at the major ERα binding site 100kb upstream of the *CCND1* promoter. This site also gains p300 occupancy and chromatin accessibility (ATAC-seq) upon addition of E2 and we assume this to be an E2-dependent enhancer (S5A Fig). DNA FISH revealed significantly increased enhancer-CCND1 promoter inter-probe distances after E2 addition. No significant changes were seen between enhancer-control probes (S5C and S5D Fig and S2 Table).

## E2-induced changes in enhancer-promoter contact frequency, and spatial proximity, are enhancer-dependent

To ascertain whether E2-induced changes in chromosome conformation and spatial organization are dependent on specific ERα enhancer activation, we performed C-TALE and DNA FISH on the cell clones deleted for either the *GREB1* or the *NRIP1* enhancers that we show are required for correct E2-dependent gene activation (Figs 1D and S2). The size of the engineered deletions—around 600 bp—is smaller than both the bin size used in our Hi-C analysis and the region covered by the DNA FISH probes, making it possible to look at contact frequencies and spatial distances involving these deletion-harbouring regions.

At both *GREB1* and *NRIP1* loci, C-TALE shows that E2-triggered increased enhancer-promoter contact frequencies are lost upon enhancer deletion (Figs 2E and S6A and S6B arrowheads). Even though E2 induced *GREB1* transcription is not totally impaired by enhancer deletion, the impact of the decreased transcriptional activity on E2 dependent 3D chromatin structure within the *GREB1* transcribed unit is also seen in the heatmaps (arrowed in S6A Fig). DNA FISH shows that the E2 triggered increase in enhancer-promoter inter-probe distances is also lost in the enhancer-deleted cells at both studied loci (Figs 2F, S6C and S2 Table). The deletions had no impact on spatial distances measured between the enhancers and control probes (S6D Fig and S2 Table). We conclude that both E2-dependent contact frequencies, and enhancer-promoter spatial separation, are affected by the enhancer deletions but in opposite directions–a loss of enhancer-promoter contact in chromosome conformation capture assays and decreased enhancer-promoter spatial separation assayed by DNA FISH.

## Enhancer-promoter contacts and spatial proximity are ligand dependent

Our data implicate liganded (E2) ERα binding in inducing enhanced enhancer-promoter "contact" and yet decreased spatial proximity. ERα liganded with Tamoxifen (4OH)—a selective estrogen receptor modulator—has been reported to occupy broadly the same genomic sites as E2-ERα, including at the *GREB1* and *NRIP1* enhancers, albeit with reduced occupancy relative to E2 [23,32]. We examined ERα recruitment to the promoters and enhancers of

GREB1 and NRIP1 after either E2 or 4OH treatment by chromatin-immunoprecipitation-PCR (ChIP-PCR) (Fig 3A and S3 Table). As expected, ERα occupancy at the enhancers, and to a lesser extent promoters, of *GREB1* and *NRIP1* is stimulated by E2. Relative to E2-treated cells, reduced ERα occupancy at the enhancers, but not the promoters, of *GREB1* and *NRIP1* is detected in the presence of 4OH.

Rather than the co-activators recruited by E2-ERα, 4OH bound ERα recruits co-repressors thereby repressing the hormonal response [23,33]. Consistent with this, ChIP-PCR indicates reduced RNA polymerase II (RPolII) occupancy at the promoters of *GREB1* and *NRIP1* in the presence of 4OH compared with E2-treated cells, albeit this doesn't reach statistical significance for *NRIP1* (Fig 3B and S3 Table). Statistically significant changes in RPolII occupancy at the enhancers were not detected. Concordant with reduced RpolII at the gene promoters, RT-qPCR showed that 4OH impedes transcriptional activation of E2-target genes. No induction of *GREB1* mRNA is detected by 4OH treatment (Fig 3C and S3 Table). A low level of 4OH-induced *NRIP1* expression is detected, but significantly lower than that seen with E2.

At the *GREB1* locus the enhancer-promoter contact frequencies induced by E2 (Fig 2) are not seen with 4OH (arrowheads in Figs 3D, 3E and S7A and S7B). Consistent with the absence of transcriptional induction in response to 4OH, the increased short-range contact frequencies along the length of the *GREB1* gene, that are induced in response to E2 (Fig 2), are also absent in 4OH-treated cells (arrowed in Figs 3D and S7A). Relative to E2 treated cells, enhancer-promoter contact frequencies at *NRIP1* are also diminished in response to 4OH (Figs 3D, 3E and S7A and S7B).

Unlike with E2, 4OH also does not induce any significant change in *GREB1* or *NRIP1* enhancer-promoter distances detected by DNA FISH, compared with vehicle treated cells (Figs 3F,3G, S7C and S3 Table). No changes in spatial distances between the *GREB1* or *NRIP1* enhancers and control probes are detected under any of the conditions (S7D Fig and S3 Table). We conclude that both enhancer-promoter contact frequencies and enhancer-promoter spatial proximity reflect the levels of ERα recruitment to enhancers and the nature of the co-factors and transcriptional machinery recruited at target loci thereafter.

## Spatial proximity, but not enhancer-promoter contacts, depend on transcription

To investigate the role of the transcriptional machinery in enhancer-promoter contacts and spatial proximity, we analysed the effects of RNA polymerase II (RPolII) inhibition on E2-induced changes in 3D chromosome conformation.

Flavopiridol (FLV) inhibits several CDKs including CDK9/PTEF-b, RNA polymerase II Ser2 CTD phosphorylation and transcription elongation, whereas triptolide (TRP) inhibits transcription initiation via TFIIH and leads to degradation of RPolII [34]. We treated hormone starved MCF-7 cells with FLV or TRP for 5 min before adding E2 for 30 min (Fig 4A). In this time frame we do not expect significant reduction of global RpolII levels [35,36].

It was reported that ERα, its co-activators, and the transcription machinery are still recruited to enhancers in the presence of flavopiridol, even though transcription of E2 target genes and eRNAs is inhibited [18]. Indeed, immunofluorescence still detected E2-dependant association of ERα with chromatin in the presence of either FLV or TRP, though this is diminished in the case of triptolide (S8A Fig). By ChIP-PCR we still detected robust E2-dependent ERα recruitment to the enhancers of *GREB1* and *NRIP1* in the presence of FLV (Fig 4B and S4 Table). ERα recruitment could also be detected in the presence of TRP, though this was somewhat blunted. ChIP-PCR analysis of ERα occupancy at the gene promoters was inconclusive. As expected for the different modes of action of the two inhibitors, E2-dependent RpolII

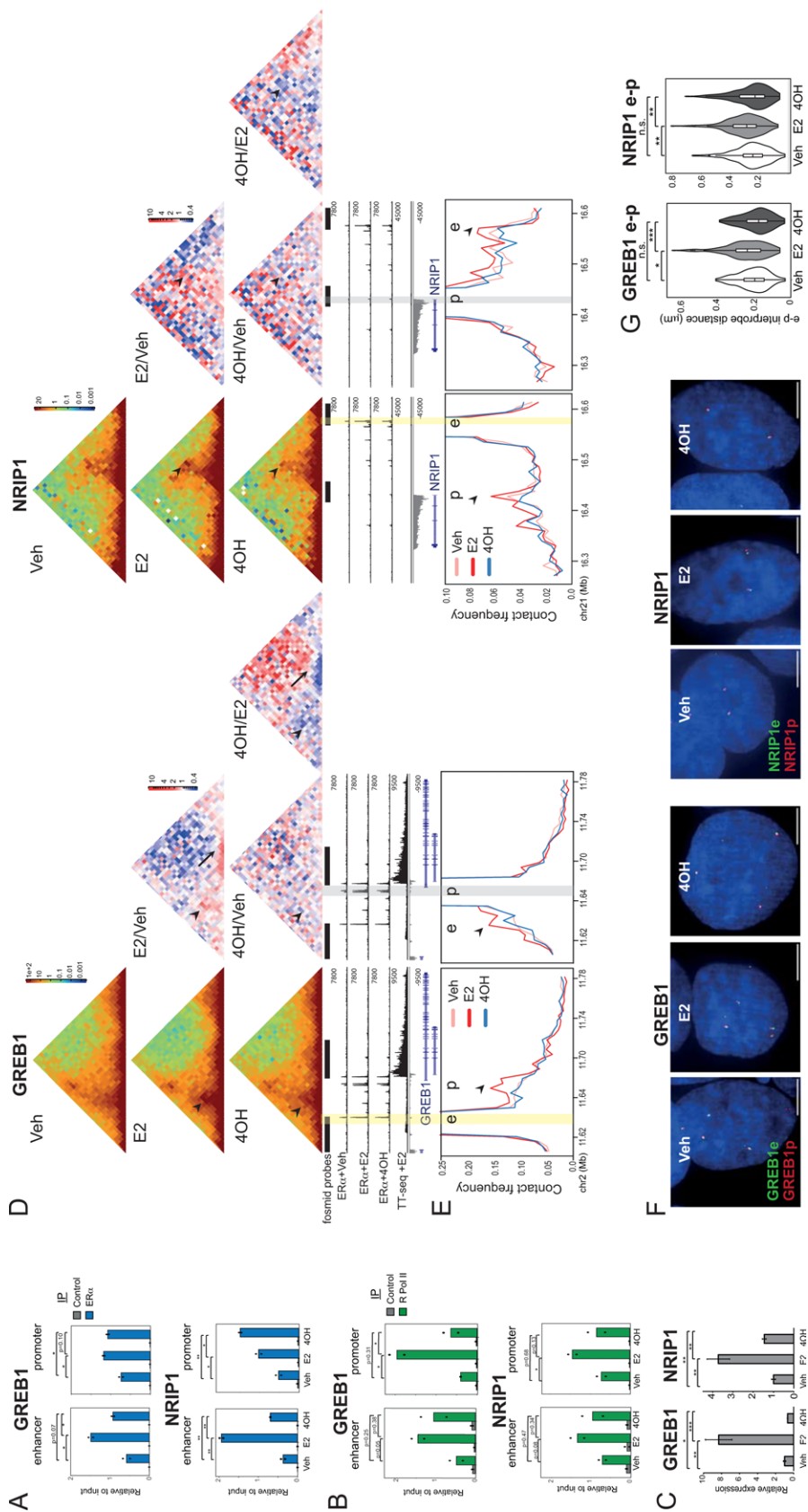

**Fig 3. Enhancer-promoter interaction frequencies and spatial separation depend on the ERα ligand.** A) ChIP-qPCR for ERα at the indicated regions in MCF-7 cells treated with vehicle (veh), E2 or tamoxifen (4OH) for 30 min. T-test, mean of 2 biological replicates shown as dots. B) As in (A), but for at RpolII occupancy. C) RT-qPCR of *NRIP1* and *GREB1* pre-mRNAs in MCF-7 cells treated with veh, E2 or 4OH for 1h. T-test, Bonferroni correction for multiple testing, mean +/- SD of 3 biological replicates. D) C-TALE heatmaps at *GREB1* (left; 5kb resolution) and *NRIP1* (right; 10kb resolution) loci from MCF-7 cells treated with veh, E2 or 4OH for 30 min. ERα +Veh, ERα +E2, ERα +4OH ChIP-seq data [23] and TT-seq +E2 30 min tracks are shown. Enhancer and promoter regions highlighted with yellow and grey bars, respectively. In (left) red-green-blue heatmaps, each pixel represents the normalized contact frequency between a pair of loci. In the red-white-blue heatmaps to the right each pixel represents the ratio between 4OH-treated samples and veh- or E2-treated samples. Arrowheads indicate pixels corresponding to enhancer-promoter interaction frequencies. Arrows indicate contacts across the *GREB1* gene body. E) Virtual 4C plots derived from normalized contact frequencies shown in (D) and using the enhancer (e, left) and promoter (p, right) regions as viewpoints. Arrowheads indicate the promoters (left) and the enhancers (right). Genome coordinates: hg19. Data for a biological replicate are in S7A and S7B Fig. F) Representative images of nuclei (DAPI, blue) from cells treated with veh, E2 or 4OH and hybridised with fosmid probes targeted to the enhancer (e, green) and promoter (p, red) regions of *GREB1* or *NRIP1*. Scale bar, 5µm. G) Violin plots showing the distribution of DNA FISH interprobe distances in cells treated with veh, E2 or 4OH using e-p fosmid probes at *GREB1* or *NRIP1* loci. Boxes indicate median and interquartile distances. Two-sided Mann-Whitney test, Holm-Bonferroni correction for multiple testing. n.s. p>0.05, **p<0.01, ***p<0.001, ****p<0.0001. Data from a biological replicate are shown in S7C Fig. Statistical data for Fig 3 are in S3 Table.

recruitment was still detected at the *GREB1* and *NRIP1* promoters in the presence of FLV, though this did not reach p<0.05 at the *NRIP1* promoter even for vehicle treated cells (Fig 4C and S4 Table). In contrast, no RpolII was detected at the gene promoters in triptolide-treated cells in the presence of E2. The same trend was seen for RpolII occupancy at the *GREB1* and *NRIP1* enhancers, but the data did not reach statistical significance.

As expected, both transcriptional inhibitors abolished the E2-dependent induction of *GREB1* and *NRIP1* mRNAs (Fig 4D and S4 Table).

Despite the absence of transcriptional activation, and the absence of promoter bound RpolII in the case of TRP-treated cells, E2 still induced increased C-TALE enhancer-promoter contact frequencies at both *GREB1* and *NRIP1* in the presence of either FLV or TRP (Figs 4E, 4F arrowhead and S8B and S8C). Consistent with the inhibition of transcription, both inhibitors blocked the E2-induced gain of short-range contacts across the *GREB1* gene body.

In contrast, flavopiridol and triptolide abolished the E2-induced loss of enhancer-promoter spatial proximity at *GREB1* and *NRIP1*, as assayed by DNA FISH (Figs 4G, 4H, S8D and S4 Table). There was no effect of E2 on the distances between control probes under all conditions (S8E Fig and S4 Table). The use of two RPolII inhibitors therefore allowed us to separate the effects of E2 addition on 3D genome organization as assayed by chromosome conformation capture vs DNA FISH: while enhancer-promoter contact frequencies assayed by C-TALE are not affected by these transcription inhibitors, the enhancer-promoter spatial separation observed by DNA FISH is transcription-dependent.

## Discussion

Chromosome conformation capture and 3D DNA FISH are linchpins methodologies for 3D genome investigation. For the most part data from these orthogonal approaches are congruent, but in a few cases spatial distances measured by DNA FISH and proximity ligation frequencies from chromosome conformation capture between the same loci are not so readily reconciled [3,6].

One of the most intensively studied areas of 3D genome organization is the relationship between enhancers and their target gene promoters. Whereas high-resolution chromosome conformation capture studies show enriched enhancer-promoter contact-frequencies [8,9], there are several examples where live-cell imaging and 3D DNA FISH in mammalian cells do not show the frequent enhancer-promoter proximity (<200nm) that might be expected from

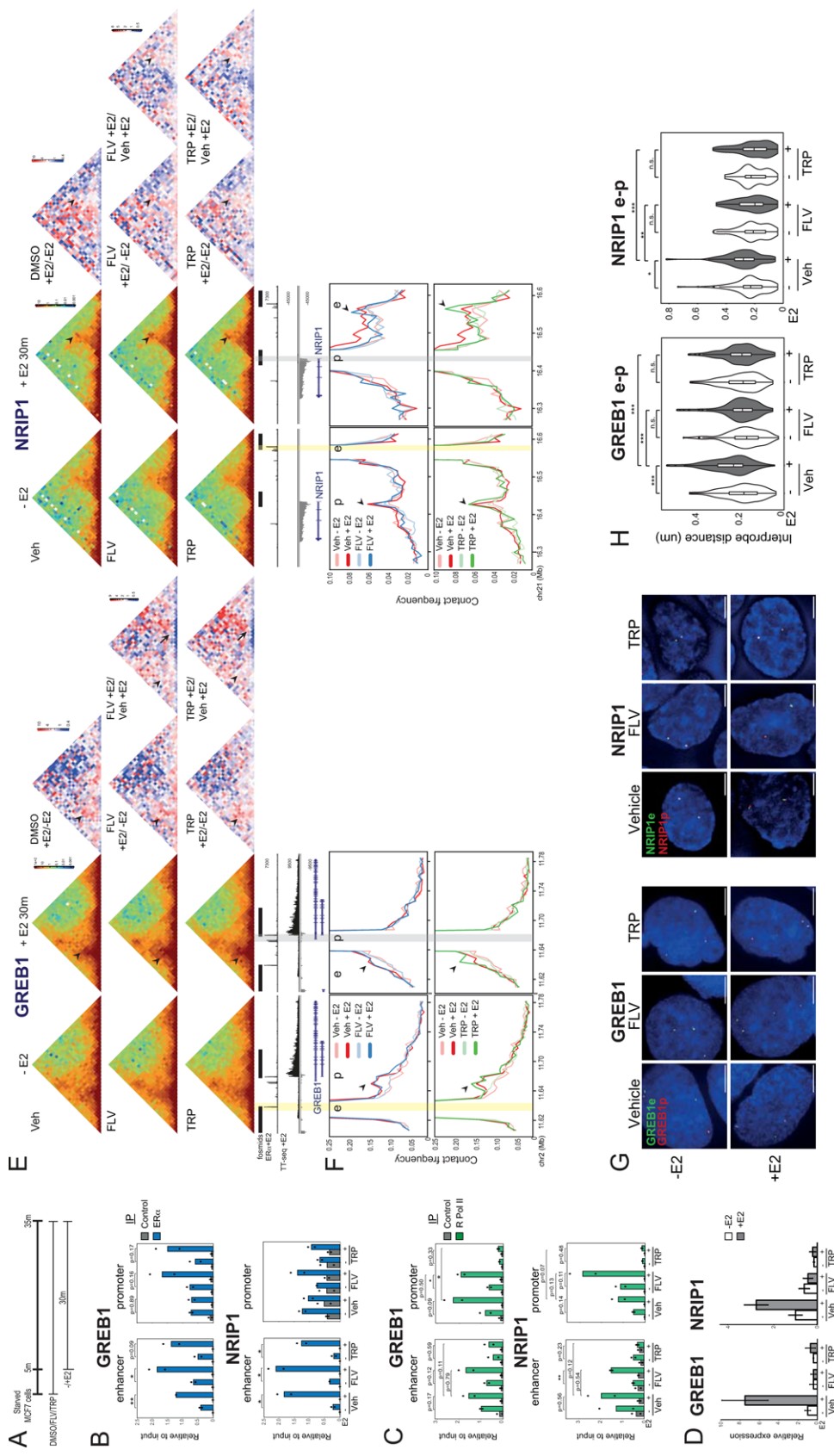

**Fig 4. Inhibition of transcription impacts E2-induced enhancer-promoter spatial separation, but not chromosome conformation capture contact frequencies.** A) Top; Schematic showing the treatment time-points. B) ChIP-qPCR for ERα at the indicated regions in MCF-7 cells treated with vehicle (veh), flavopiridol (FLV) or triptolide (TRP) for 5 min to prior to E2 treatment (+E2) 1h. T-test, mean of 2 biological replicates shown as dots. C) As in (B), but for at RpolII occupancy. D) RT-qPCR of *NRIP1* and *GREB1* pre-mRNAs in MCF-7 cells treated with vehicle, flavopiridol (FLV) or triptolide (TRP) for 5 min prior to E2 treatment (+E2) for 1h. Means +/- SD of 3 replicates. E) Capture C-TALE-C heatmaps at *GREB1* (5kb resolution) and *NRIP1* (10kb resolution) loci from MCF-7 cells treated with vehicle, FLV or TRP for 5 min prior treatment without (-E2) or with E2 (+E2) for 30 min. In the middle red-white-blue heatmaps each pixel represents the ratio between +E2 and–E2 treated samples for vehicle, FLV or TRP treted samples. To the right, red-white-blue heatmaps each pixel represents the ratio between FLV +E2 or TRP +E2, and vehicle +E2 treated samples. Arrowheads indicate pixels corresponding to enhancer-promoter interaction frequencies. Arrowsindicate contacts along the *GREB1* gene body. Data from an independent biological replicate are in S8B Fig. F) Virtual 4C plots derived from the normalized contact frequencies shown in (E) using the enhancer (left) and promoter (right) regions as viewpoints. Genome coordinates: hg19. Arrow heads indicate the promoters (left) and enhancer (right). G) Representative images of nuclei (DAPI, blue) of MCF-7 cells treated as in (A) showing the DNA FISH signal from fosmid probes targeted to the enhancer (e, green) and promoter (p, red) regions of either GREB1 or NRIP1 loci. Scale bars, 5μm. H). Violin plots showing the distribution of DNA FISH interprobe distances of cells treated as in (A), using e-p fosmid probes for either GREB1 or NRIP1 loci. Boxplots indicate median distances. Two-sided Mann-Whitney test, Holm-Bonferroni correction for multiple testing. Data from a biological replicate are in S8D Fig. n.s. p>0.05, *p<0.05, **p<0.01, ***p<0.001. Statistical data for Fig 4 are in S4 Table.

the results of proximity ligation-based assays [12–14]. Contact-independent models are emerging that require general spatial enhancer-promoter proximity but not protracted enhancer-promoter direct contact for the activation of transcription [2,15,37].

Inducible systems are key to understanding how enhancer-promoter proximity is linked to gene activation. In mESCs, synthetic transcription factors have been used to study enhancer-driven gene activation unexpectedly revealing decreased enhancer-promoter spatial proximity upon enhancer activation [13,15], and inconsistent with an enhancer-promoter direct contact looping model [2]. However, both the efficiency of transcription and the temporal resolution in these systems are low, such that events occurring immediately downstream of enhancer activation, and the genome organization specific to transcribing alleles, may have been missed.

Here we address these limitations using a system of rapid and efficient enhancer-dependent gene activation in MCF-7 cells mediated by the binding of liganded ERα at the enhancers of two well studied target genes–*GREB1* and *NRIP1*. Upon hormone stimulation, ERα rapidly binds to the enhancers, which are already in an open chromatin state and bound by pioneer transcription factors [20,38]. This ensures rapid and highly efficient transcriptional induction of target genes–nascent RNA FISH showed that the majority of alleles are in the process of being transcribed 20–60 mins after E2 addition (Fig 1). Data on 3D genome organization assayed in the E2-stimulated state is therefore chiefly attributable to actively transcribing alleles.

Using Hi-C coupled to target enrichment (C-TALE) at the two loci we detected the increased enhancer-promoter contact frequencies upon hormone stimulation expected from enhancer-promoter looping models. However, DNA FISH reveals a loss of enhancer-promoter spatial proximity at these same time points (Fig 2), consistent with what we have seen at a developmental locus [13,15], and at target loci of another nuclear hormone receptor–the glucocorticoid receptor [30]. We note that this contrasts with live cell imaging data in Drosophila embryos in which enhancer-driven transcription of a reporter gene corresponds to more sustained spatial proximity between the reporter and enhancer loci [11].

We used genetic and biochemical perturbations to dissect the molecular events driving these apparently opposing observations. Both the E2-stimulated increased enhancer-promoter C-TALE contact frequencies, and the decreased spatial proximity detected by DNA FISH, depend on the presence of the enhancers (Fig 2). This contrasts with the lack of significant enhancer-promoter distance changes measured at the *Sox2* locus in mESCs when the SCR enhancer is mutated compared to wild-type cells [14].

Both E2-stimulated increased enhancer-promoter C-TALE contact frequencies, and the reduced spatial proximity, also depend on occupancy levels of liganded ERα at the enhancers and the subsequent recruitement of the transcription machinery, since both are lost in the presence of the ERα antagonist tamoxifen (Fig 3). Finally, the use of transcription inhibitors allowed us to further disentangle the drivers of enhancer-promoter contact frequencies and spatial proximity. While E2-dependent enhancer-promoter contact frequencies persist when transcription initiation or elongation are blocked with triptolide or flavopiridol, the E2-dependent reduction of enhancer-promoter proximity seen by DNA FISH is abrogated (Fig 4). In the time frame of the experiments performed here, we show that ERα is still recruited to GREB1 and NRIP1 enhancers by E2 in the presence of these inhibitors. E2 still induces RNA polymerase II recruitment to the promoters of these two gene in the presence of flavopiridol, but not in the presence of triptolide. It was reported that transcription inhibition has little impact on higher order 3D chromatin organization in mESCs [9] while enhancer-promoter contact frequencies were shown to be somewhat reduced [9,39]. It should be noted however, that this reduction is only noticeable at the genome wide level and that the foci of these interactions were largely unaffected [9,39]. Our observations also appear to differ from what has been seen at the Sox2-SCR locus in mESCs where triptolide has no affect on measured enhancer-promoter distances [14]. However a notable difference is that whereas *Sox2* is being actively transcribed in mESCs, here we are looking at the acute transcriptional induction of genes from an inactive state. We conclude that, at the two loci we have examined in detail, the gain of enhancer-promoter contact frequencies detected by Hi-C/C-TALE coincides with E2-liganded ERα recruitment to enhancers, whereas decreased enhancer-promoter proximity corresponds with E2-dependent transcription. Most noticeably in the case of *NRIP1*, increased enhancer-promoter separation occurs as soon as 5 min after E2 stimulation. This is before an induction of nascent transcription of the gene is detected by either TT-seq or RNA FISH (Fig 1). However some, albeit not statistically significant, induction of eRNAs is detected at this early time-point (S1 Fig).

We also detected an enrichment of short-range, and depletion of longer-range, C-TALE interactions along gene bodies linked to E2-induced transcription. These depend on transcription, being lost in the presence of tamoxifen or transcription inhibitors. This is most noticeable in the case of *GREB1*, a long gene with approximately 30 exons, in comparison with *NRIP1's* simple gene structure (up to 4 exons). These observations are compatible with a model in which genes stiffen and decompact as the transcription machinery elongates through, and nascent RNAs and RNA binding proteins coat, the gene bodies constraining their ability to make long-range contacts [40].

At the cell population level, we observe increased chromosome conformation capture contact frequencies between regions that, at the same time, appear to become further apart, on average, within the nucleus. Whilst imaging data is collected from almost every allele in the few hundred cells that are imaged per experiment, it is not known what proportion of alleles in the millions of cells that go into the Hi-C reaction contribute to the enhancer-promoter contact frequencies detected. We therefore cannot exclude that there is very transient physical contact between the enhancers and promoters that is too infrequent to detect by DNA FISH, but that is captured by C-TALE. However, the fact that most of the alleles we visualise in this system are being actively transcribed means that such a "kiss and run" contact is not required for each burst of transcription. We also acknowledge the technical limitations of imaging in fixed cells, including localisation error and probe size, that may tend to over-estimate inter-probe distances [41]. Indeed we note the smaller inter-probe distances we measured using small (10kb) DNA FISH probes compared with larger fosmid probes at the same locus. Nevertheless, this cannot account for the lower cumulative frequency of close enhancer-promoter

distances we observe upon E2-induced activation at E2 target loci compared to the pre-induced state, and that is not seen at control loci.

Both 3C techniques and DNA FISH rely on protein-protein and DNA-protein crosslinking with formaldehyde, which is widely used and yet incompletely understood [42]. The recruitment of ERα, co-activators and the transcriptional machinery results in megadaltons of potentially additional cross-linkable protein complexes at ER-responsive enhancers following E2, but not 4OH, treatment. Crosslinking between somewhat distant genomic regions is also plausible via a crosslinked meshwork of macromolecules that bridge between two loci. This cross-linking radius is thought to depend on protein occupancy, the abundance of specific amino acids and crosslinking efficiency [43,44]. Indeed, we note that the physical enhancer-promoter distances we measure upon E2 stimulation might fall within estimated 3C crosslinking and ligation radii [3,41,44].

Formaldehyde crosslinking is also affected by protein dynamics. Protein cross-linking to DNA requires a long residence time (many seconds) [45,46] and proteins can artifactually concentrate into large puncta, or can appear to leave foci, during fixation if the fixation rate is slower than protein diffusion rates [47]. E2-dependent increased enhancer-promoter contact frequencies detected by chromosome conformation capture could therefore be influenced by the residence time and exchange rates of the large multicomponent protein complexes recruited by ERα (Fig 5). In contrast, the loss of enhancer-promoter spatial proximity detected by DNA FISH appears to be linked to the production of RNAs (eRNA and mRNA) at the target loci, perhaps due to the effect of these RNAs on biomolecular condensate formation [48].

Our work reinforces that high resolution enhancer-promoter contact frequencies derived from 3C techniques cannot always be simply interpreted as physical distances. Careful consideration, including study of other loci and in different systems, needs to be given to how dynamics and the functional, biochemical and biophysical environments of genomic loci in different biological systems and under different conditions may impact the data obtained from imaging and chromosome conformation assays.

## Methods

### Cell culture and treatments

MCF-7 cells were grown in Dulbecco's Modified Eagle's medium (DMEM, GIBCO021969-035), high glucose and supplemented with 2 mM L-glutamine, 100 units/ml penicillin, 6.5 μg/ml streptomycin and 10% foetal calf serum (FCS, Gibco 10270).

Before hormone stimulation, cells were seeded in growth media and 24hr later transferred into starving media comprising phenol-red free DMEM, penicillin/ streptomycin and 10% charcoal-stripped FCS for 96hr. FCS was charcoal-stripped of steroid hormones by heat inactivation in a waterbath at 56°C for 30 minutes (mins) before addition of 2000U/l sulfatase (Sigma S9626-5KU, resuspended in 0.2% NaCl to 1000 U/ml). After 2 hours (hrs) incubation at 37°C, the pH was adjusted to 4.2 using HCl and a charcoal mix (for 1 litre: 5 g charcoal (Sigma-Aldrich 05105), 25 mg dextran T70 (USP, 1179741), 50 ml water) was added and incubated overnight (o/n) at 4°C with stirring. Charcoal was removed by centrifugation at 500 $g$ for 30 mins at 4°C, the pH re-adjusted to 4.2 and a second charcoal mix added, incubated and then removed. Centrifugation was repeated to remove residual charcoal and the pH adjusted to 7.2 with NaOH. Stripped FCS was filter sterilized, aliquoted and stored at -20°C.

Cells were transferred into phenol-red free DMEM supplemented with either 10 nM 17ß-estradiol (Sigma E2758) (+E2) or ethanol (-E2) for the indicated time points. 10μM Tamoxifen (Sigma T3652) (4OH) was used. For transcription inhibition, media was supplemented with either DMSO (vehicle), 10 μM Flavopiridol (Cambridge Bioscience 2090–2) or 10 μM

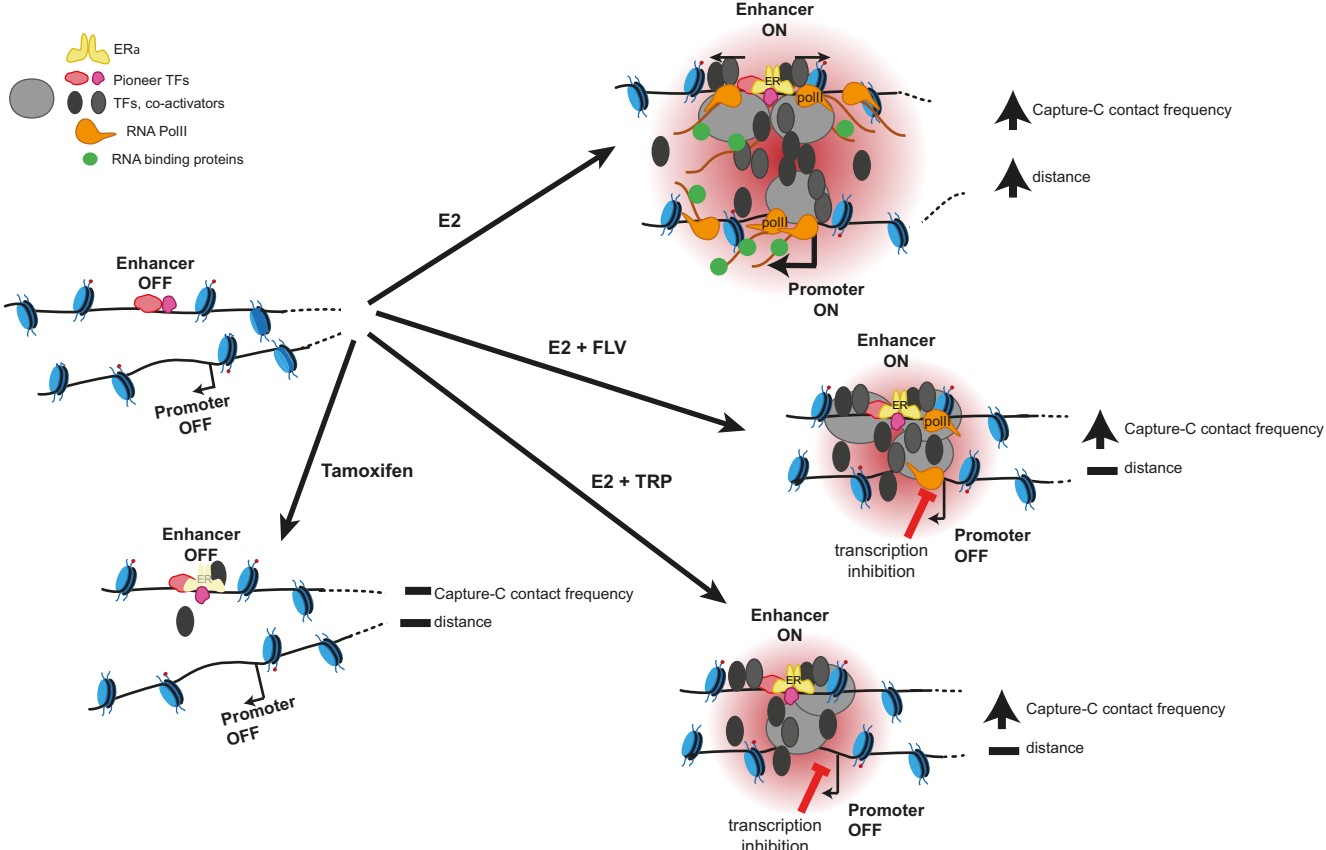

**Fig 5. Model for enhancer-promoter communication leading to increased chromosome conformation capture contact frequencies but loss of spatial proximity during transcriptional activation.** Upon estradiol (E2) stimulation, ERα binds to enhancers recruiting co-activators and the transcription machinery, and leading to the transcription of eRNAs and target mRNAs. This results in increased enhancer-promoter contact frequencies and yet increased spatial separation of enhancer and promoter (top right). When transcription is inhibited with flavopiridol (FLV) during E2 stimulation, ERα, co-activators and the transcription machinery are still recruited but without the transcription of eRNAs or mRNAs. In the presence of triptolide (TRP) ERα is still bound at enhancers, but RNA polymerase II is nor recruited. Both FLV and TRP still result in increased enhancer-promoter contact frequencies in response to E2 addition, but without changes in spatial distance (middle right). With tamoxifen as ligand, ERα occupancy is reduced and there is no activation of the target genes. In this context, neither enhancer-promoter contact frequencies nor spatial distances change in comparison with the unstimulated situation (bottom right). We speculate that the increase in enhancer-promoter contact frequencies observed upon hormone stimulation is driven by the recruitment of large multicomponent protein compexes while increased enhancer-promoter spatial separation seems to linked to the molecular events associated with transcription.

Triptolide (Sigma T3652). After 5 mins, 1/30 vol of media was incorporated with either ethanol (-E2) or 17ß-estradiol (+E2) to a final concentration of 10 nM for 30 mins.

## TT-seq

TT-sequencing was adapted from [5]. $3 \times 10^6$ MCF-7 cells were seeded in T75 flasks, starved, and then treated with E2 or vehicle for the indicated time points. For every time point, 4-thiouridine (4SU; Sigma T4509) was added to a final concentration of 500 μM for 5 min before cells were harvested with 2ml/flask of Trizol (Invitrogen 15596026). RNA was prepared as previously described [5]. RNA samples were sonicated with a Diagenode Bioruptor Plus for 30 s at a high setting. 10–100 ug of RNA were biotinylated, uncoupled biotin was removed and biotin-labelled RNA purified as previously descried [5]. Libraries were constructed using the NEBNext Ultra II directional RNA library preparation kit for Illumina according to the protocol for purified mRNA or ribosome-depleted RNA and with a 5-min RNA fragmentation step

(NEB E7760). Library PCRs were supplemented with 2× SYBR dye (Sigma S9430) to monitor amplification by quantitative(q)PCR on a Roche LightCycler 480. To allow for sample multiplexing, PCRs were performed using index primers (NEBNext multiplex oligos for Illumina, set 1, E7335) and amplified to linear phase. Libraries were combined as an equimolar pool and sequenced on an Illumina NextSeq on a single high-output flow cell (paired-end 75-bp reads).

## TT-seq data analysis

For each demultiplexed sample, multiple raw Fastq files were merged (individually for reads 1 and 2). Adapter sequences were removed using Cutadapt v1.16 in paired end mode (options: -q 20 -a AGATCGGAAGAGCACACGTCTGAACTCCAGTCA -A AGATCGGAAGAG CGTCGTGTAGGGAAAGAGTGT -m 20). Adapter-trimmed Fastq files were aligned to the human genome (hg19) using STAR v2.7.8 for paired end sequence data and using Ensemble GRCCh37gene annotation to generate BAM files (options:—outFilterType BySJout—outFilterMultimapNmax 20—outSAMunmapped Within). Bam files were indexed using samtools (v1.3) index; bigwig files used for data visualization on UCSC genome browser, one file per strand, were generated using deeptools (v3.5.1) bamCoverage (options:—normalizeUsing RPKM -bs 1—filterRNAstrand forward or reverse). Replicates were merged in single bigwig file using deeptools bigwigCompare using the mean operation. Read counts over specific genomic regions were computed using featureCounts v2.0.1 (options: -p -Q 10 -s 2). To compute read counts over TSSs, regions 1kb downstream of TSSs of Ensemble GRCCh37gene annotation were used. To compute read counts over intergenic regions, pooled ATAC-seq peaks [23] at least 1kb away from annotated TSSs and extended 1.5 kb at each side, were used. Differential expression analysis was performed using edgeR on R.

## ChiP-seq data analysis

Data from [16] (ArrayExpress: E-MTAB-785), [49] (ArrayExpress: E-TABM-828) and [23] (NCBI GEO: GSE117943) were re-analyzed. Fastq files were quality assessed (FastQC v0.11.4), adapters trimmed (Cutadapt v1.16) and aligned to the human genome (hg19) using Bowtie2 v2.2.6 in either single- or paired-end modeto generate sam files (options for paired-end:—no-discordant—no-mixed—no-unal—very-sensitive -X 2000). Potential PCR duplicates were removed using samtools (v1.3) rmdup. Bigwig files used for data visualization on the UCSC genome browser were generated using deeptools (v3.5.1) bamCoverage (options:—normalizeUsing RPKM -bs 1). Replicates were merged in single bigwig file using deeptools bigwigCompare using the mean operation. Peaks were called using MACS2 (v2.1.1) callpeak (options: -g hs -B -q 0.01) for each replicate. Common peaks for each treatment were obtained using bedtools (v2.27.1) intersect.

## CRISPR deletions

pSpCas9(BB)-2A-GFP (PX458, Addgene 48138) plasmids including either the left or right gRNA (S5 Table) were co-transfected into MCF-7 cells using Lipofectamine 3000 (Invitrogen, L3000015). After 24 hrs, single GFP+ve cells were seeded into 96 well plates. Cell lysates from different clones were obtained using DirectPCR Lysis Reagent (Viagen Biotech, 102-T) and genotyping was performed by PCR using the primers in S5 Table. PCR bands were cloned into the TOPO-TA cloning kit (Invitrogen, 450071) and validated by Sanger sequencing.

## RNA FISH

Custom Stellaris RNA FISH probes were designed against *GREB1* and *NRIP1* nascent mRNA first introns (pool of 48 unique 22-base polymer probes) using the Stellaris RNA FISH Probe Designer (www.biosearchtech.com/stellarisdesigner (version 4.2)). $4x10^5$ MCF-7 cells were seeded on Thermo Scientific SuperFrost Plus Adhesion slides (15438060), starved and treated as specified. Cells were fixed with 4% paraformaldehyde (pFa) in PBS for 10 min, rinsed with PBS 3 x for 2 min and permeabilized with 0.5% Triton X-100/ PBS for 10 min. Following 3 x 2 min rinses, slides were incubated in wash buffer (2× SSC, 10% deionized formamide) for 5 mins at r.t. Slides were hybridized with *GREB1* or *NRIP1* Stellaris FISH Probe sets labeled with Quasar 670 (Biosearch Technologies) following the manufacturer's instructions (www.biosearchtech.com/stellarisprotocols). RNA FISH probes were hybridised to slides as previously described [15].

## 3D DNA FISH

**Probe preparation.** For fosmid probes (S6 Table), 1 µg of fosmid DNA was labeled by nick translation to incorporate biotin-dUTP (Roche 11093070910) or digoxigenin-dUTP (Roche 11093088910) and prepared for hybridization as previously described [5]. For 10kb PCR probes (S6 Table), fragments generated by PCR from BAC templates (S7 Table) were cloned into TOPO-TA plasmids (Invitrogen). 1 µg of plasmid DNA was labeled by nick translation to incorporate biotin-dUTP (Roche 11093070910), digoxigenin-dUTP (Roche 11093088910) or Green496-dUTP-labeled (Enzo Life Sciences) and used as above.

**Cell fixing, denaturing and hybridization.** Cells were seeded, starved, treated, fixed and permeabilized as for RNA FISH. Following permeabilization and rinsing, slides were air dried and stored at -80˚C till use. Slides were incubated in 100 µg/mL RNase A in 2× SSC for 1 hr at 37˚C, washed briefly in 2× SSC, passed through an ethanol series (70, 90 and 100%, 2 min in each), and air-dried. Slides were incubated for 5 min in a dry incubator at 70˚C, denatured in 70% formamide/2× SSC (pH 7.5) for 40 min at 80˚C, cooled in 70% ethanol for 2 min on ice, and dehydrated by immersion in 90% ethanol for 2 min and 100% ethanol for 2 min prior to air drying. Probes in hybridization mix were added to the slides and incubated o/n at 37˚C. Following a series of washes in 2× SSC (45˚C) and 0.1× SSC (60˚C), slides were blocked in blocking buffer (4× SSC, 5% Marvel) for 5 mins. For 2-color detection (either with fosmid or 10 kb derived probes), the following antibody dilutions were made: anti-digoxigenin-Rhodamine FAB fragments (Roche, cat. no. 11207750910) 1:20; Texas Red anti-sheep 1:100 (Abcam ab6745) and Avidin-FITC1:500 (Vector Laboratories A-2011); biotinylated anti-avidin (Vector Laboratories, cat. no. BA-0300) 1:100, and Avidin-FITC 1:500. For 3-color detection the following antibody dilutions were made: anti-digoxigenin (Roche, cat. 11333089001) 1:10; AF647 anti-sheep (Invitrogen, cat. A-21448) 1:10 and Texas Red avidin (Vector Laboratories, cat. no. A-2016) 1:500; biotinylated anti-avidin (Vector Laboratories) 1:100; Texas Red avidin (Vector Laboratories, cat. no. A-2016) 1:500. Slides were incubated with antibody in a humidified chamber at 37˚C for 30–60 mins in the following order, with 4× SSC/0.1% Tween 20 washes in between: fluorescein anti-digoxigenin, fluorescein anti-sheep, biotinylated anti-avidin, streptavidin-Cy5 for 3-color detection; Texas Red avidin, biotinylated anti-avidin, Texas Red avidin for 2-color detection. Slides were treated with a 1:1,000 dilution of DAPI (stock 50 µg/mL) for 5 mins before mounting in Vectashield.

## Image acquisition and deconvolution

Slides from RNA and DNA FISH using fosmid derived probes were imaged on an epifluorescence microscope as previously described [5]. Step size for z stacks was set to 0.2 µm. Hardware

control and image capture were performed using Nikon Nis-Elements software (Nikon) and images were deconvolved using a calculated PSF with the constrained iterative algorithm in Volocity (PerkinElmer). RNA FISH signal quantification was carried out using the quantitation module of Volocity (PerkinElmer). Number of expressing alleles was calculated by segmenting the hybridization signals and scoring each nucleus as containing 0, 1 or 2 RNA signals. DNA FISH measurements were made using the quantitation module of Volocity (PerkinElmer) and only alleles with single probe signals were analysed to eliminate the possibility of measuring sister chromatids.

Slides for DNA FISH using 10kb probes were imaged on a SoRa spinning disk confocal microscope (Nikon CSU-W1 SoRa) in 0.1 μm steps. Images were denoised and deconvolved using NIS deconvolution software (blind preset) (Nikon) and DNA FISH measurements were as for fosmid probes.

## C-TALE

The C-TALE protocol experiments was adapted from [29], with 3C libraries captured with biotin labelled fragments from BACs covering the regions of interest (S7 Table).

### 3C library preparation

$3x10^6$ MCF-7 cells were seeded in 10 cm plates, starved and treated as described above. Cells were crosslinked, lysed into nuclei and chromatin was fragmentated and *in situ* proximity ligated as decribed [29]. For chromatin fragmentation, 350 U of DpnII 50U/μl (NEB cat. R0543M) in 300 μl 1x NEBuffer DpnII (NEB cat. B0543S) were used. After de-crosslinking and DNA extraction [29] 5 μg of DNA in 500 μl sonication buffer (25 mM Tris-HCL pH8.0, 20 mM EDTA, 0.2% SDS) were sonicated using MSE Soniprep 150 Plus at medium power for two 30 sec pulses on ice to shear DNA to a size of 150–700 bp. 1 μg of the resulting DNA was used for library preparation using the NEBNext Ultra™ II DNA Library Prep Kit for Illumina (NEB cat. E7103). Library PCRs were supplemented with 1× EvaGreen (Jena Bioscience cat. PCR-379). qPCRs were performed using index primers (NEBNext multiplex oligos for Illumina, set 1, E7335) and amplified to linear phase (3–5 PCR cycles).

### Biotin-bait preparation

5 ug of pooled equimolar amounts of the selected BACs (S7 Table) were sonicated as described above for 3C libraries. Equimolarity was calculated by qPCR using primers against the chloramphenicol resistance gene (S7 Table). Forked adapters (S7 Table) were annealed in NEBuffer 2 1X (NEB cat. B7002S) to a final concentration of 20 μM in a thermal cycler set to heat at 98°C and gradually cool down to 4°C with a 1°C per 20 sec gradient.

Shredded BACs were subjected to end-preparation, A-tailing and adapter ligation as described [29]. BAC baits were PCR amplified using 5' biotinylated forward and reverse primers (S7 Table) in the following reaction mix: 10 μl 2X NEBNext Ultra™ II Q5 Master Mix (NEB cat. M0544S), 1 μl 10 μM each bio-primer, 2 μl adapter-ligated BAC DNA, 1X EvaGreen (Jena Bioscience cat. PCR-379). The number of amplification cycles (in the linear phase of the reaction), determined in a test qPCR, usually fell between 7 and 9 cycles. Around 3 μg of biotinylated baits were obtained from 25 reactions.

### Hybridization and biotin-pulldown

For each experiment, equal amounts of all libraries corresponding to each sample and amplified with different barcodes, were pooled and captured as a pool. One hybridization reaction

was done per 1 μg of library pool. Hybridization and biotin-pulldown was done as described [29].

Singly enriched 3C libraries were amplified by qPCR using p5/p7 primers (S7 Table) as described above (Biotin-bait preparation). PCR reactions were performed with the DNA still bound to the streptavidin beads. The number of amplification cycles (in the linear phase of the reaction), determined in a test qPCR, usually fell in between 13 and 16.

A second hybridization reaction, biotin pulldown and PCR amplification was performed as before. The number of PCR cycles usually fell between 7 and 10 cycles. The pool was sequenced using either an Illumina NextSeq 500 on a single midium-output flow cell (paired-end 75-bp reads) or an Illumina Nextseq 2000 using a single P1 flow cell (paired-end 75-bp reads).

## C-TALE data analysis and generation of virtual 4C profiles

C-TALE data were aligned to the hg19 genome and processed using distiller-nf 0.3.3 (https://doi.org/10.5281/zenodo.3350937) with default settings. Coolers filtered for $q \geq 30$ were used and ICE balanced [50] using the cooler [51] command 'cooler balance—cis_only—blacklist blacklist.bed' where blacklist.bed contains all regions outside of the capture probe area. Balanced contact frequencies were visualised using HiGlass in resgen.io [52]. Virtual 4C data were generated by extracting the balanced contact frequency values from all bins overlapping the viewpoint using cooltools [53] and summing them. Plots were generated using seaborn [54]. In the case of *GREB1* locus, analysis was done at a 5 kb resolution and heatmaps and plots are shown between the coordinates chr2:11602736–11790379 (hg19). In the case of *NRIP1* locus, analysis was done at a 10 kb resolution and heatmaps and plots are shown between the coordinates chr21:16263224–16630829 (hg19).

## Chromatin immunoprecipitation

Approximately $1-2 10^7$ MCF-7 cells were crosslinked for 10 min in 1% formaldehyde in PBS at room temperature. Crosslinking reaction was stopped with glycine at a final concentration of 125 mM. Cells were washed twice with cold PBS and swelled on ice for 10 min in 5 mM PIPES pH = 8, 85 mM KCl, 0.5% Igepal CA630 and 1x protease inhibitor cocktail (Roche). Following centrifugation, the pellet enriched in nuclei was resuspended in 0.5 ml sonication buffer (50 mM Tris-HCl pH8.1, 1%SDS, 10 mM EDTA pH8.0). DNA was sonicated in an ultrasonic bath (Bioruptor Diagenode) to an average length of 200–500 bp and samples were centrifuged at 15,000 x g. 20–40 μg of DNA were used in each immunoprecipitation, diluting chromatin at least ten times in 1 ml IP buffer (15 mM Tris-HCl pH8.1, 1% Triton X-100, 1 mM EDTA pH8.0, 150 mM NaCl, 1x protease inhibitor cocktail). Chromatin was immunoprecipitated o/n with 40 μl of Protein G magnetic beads (Dynabeads Protein G ThermoFisher Scientific) previously incubated with the antibody of interest for 1h at 4˚C. The following antibodies were used: 4 μg of mouse monoclonal anti-ERα F-10 (Santa Cruz Biotechnology) and 5 μg of rabbit anti-Rbp1 NTD D8L4Y (Cell Signalling)). Control immunoprecipitations were performed with no antibody. Beads were washed sequentially for 5 min each in Low-salt (20 mM Tris-HCl pH 8, 150 mM NaCl, 2 mM EDTA, 1% Triton X-100, 0.1% SDS), High-Salt (20 mM Tris-HCl pH 8, 500 mM NaCl, 2 mM EDTA, 1% Triton X-100, 0.1% SDS) and LiCl buffer (10mM Tris pH 8.0, 1mM EDTA, 250mM LiCl, 1% Igepal CA630, 1% Na-deoxycholate) for 5 min at 4˚C and then twice in TE 1X for 5 min at RT. Beads were eluted in 1% SDS and 100mM NaHCO$_3$ buffer for 1 h min at RT and crosslinking was reversed o/n after addition of NaCl to a final concentration of 200 mM and 2 μg of Proteinase K. DNA was purified with Qiaquick PCR purification columns (Qiagen) as indicated by the manufacturer. For ChIP-qPCR

immunoprecipitated DNA was analyzed with SYBR-Green real-time qPCR using the standard-curve method, qPCRs were performed using the CFX96 Touch Real-Time PCR Detection System (Bio-Rad) and the LightCycler 480 SYBR Green I Master mix (Roche 04887352001) using the program established by the manufacturer. Quantities were normalized to input. To merge different biological replicates, quantities relative to input were normalized to the average quantities within replicates and amplicons. Primers used are listed in S5 Table.

### RT-qPCR

RNA from cells grown in 6 well plates was extracted using TRIZOL (Invitrogen 15596026) and reverse transcribed using Superscirpt II Reverse Transcriptase (Invitrogen 18064022) with random decamers. qPCR was performed using the CFX96 Touch Real-Time PCR Detection System (Bio-Rad) and the LightCycler 480 SYBR Green I Master mix (Roche 04887352001) using. the program established by the manufacturer. The mean relative expression of technical replicates of each sample was measure by the CFX Manager software (Bio-Rad) according to a calibration serial dilutions curve and normalized to the mean of U2 RNA expression. Primers used are listed in S5 Table.

### Immunofluorescence

$4x10^5$ MCF-7 cells were seeded on Thermo Scientific SuperFrost Plus Adhesion slides (15438060) and treated as specified. Before fixing, slides were rinsed with 0.5% Triton X-100/ PBS for a few seconds and then fixed with pFa 4% in PBS for 10 min, rinsed with PBS 3 times for 2 min and permeabilized with 0.5% Triton X-100/ PBS for 10 min. Following 3x 2 min PBS rinses, slides were blocked with 1% BSA in PBS for 30 min at r.t. and incubated with 1:250 mouse anti ERα antibody (F-10, Santa Cruz, cat. Sc-8002) in 1% BSA in PBS for 1h. Slides were rinsed twice in PBS for 2 min and incubated for 45 min with 1:500 Goat anti-Mouse Alexa Fluor 488 antibody (Invitrogen, cat. A11001) in 1% BSA/ PBS. After rinsing, slides were incubated with a 1:1,000 dilution of DAPI (stock 50 μg/mL) for 5 mins before mounting in Vectashield (Vector Laboratories, cat. H1000). Slides were imaged on an epifluorescence microscope (Zeiss AxioImager 2, camera Hamamatsu Orca Flash 4.0 16-bit) and nuclear fluorescence intensity was quantified using CellProfiler v3.1.8.

### Statistical analyses

DNA FISH inter-probe distance data sets were compared using the two-tailed Mann-Whitney U test. Differences in DNA FISH data sets comparing categorical distributions were measured using Fisher's Exact Test. For RT-qPCR and ChIP-qPCR assays, pair-wise two-sided T-tests were used to compare the different samples. All statistical analyses were performed using R (https://www.r-project.org/).

### Supporting information

**S1 Fig. Related to Fig 1. Rapid induction of transcription at ER-responsive enhancers in MCF-7 cells.** A) Genome Browser screen shots showing ChIP-seq tracks of ERα after E2 addition and TT-seq tracks without E2 and 5, 30 and 60 min after E2 addition over the intergenic regions around the *GREB1* and *NRIP1* putative enhancers. B) Quantification of TT-seq reads over the indicated regions. These correspond to ATAC-seq peak regions extended for 1.5 kb at either side. Normalized counts per million reads (cpm) of two replicates are shown. $**p<0.01$, $***p<0.001$, $****p<0.0001$. C) Biological replicate of the data in Fig 1C. Pre-mRNA FISH for *GREB1* and *NRIP1* nascent transcripts without and with E2 in MCF-7 cells for the indicated

timepoints. The percentage of cells with 0, 1, 2, 3, 4 foci at each timepoint is shown. Two-sided Fisher exact test, *p<0.05, **p<0.01, ***p<0.001, ****p<0.0001. Statistical data for S1 Fig are in S1 Table.
(EPS)

**S2 Fig. Related to Fig 1. Enhancer deletion leads to decreased transcriptional levels with E2.** A) Diagram depicting the CRISPR-Cas9 strategy for deleting putative *GREB1* and *NRIP1* enhancer regions. The position of the gRNAs around the ERα peak and the estrogen response element (ERE), and the forward and reverse primers (blue arrows) used for screening, are shown. Genomic distances shown in bp. Sanger sequencing tracks from two independent homozygous deletion clones are shown below. Left: agarose gel of PCRs products from genomic DNA extracted from the different deletion clones using primers around the deletions. B) Biological replicate of data in Fig 1D. pre-mRNA FISH for *GREB1* and *NRIP1* nascent transcripts in WT MCF-7 cells and in cells where the respective putative ERα enhancers have been deleted. The percentage of cells with 0, 1, 2, 3 or 4 foci of each cell line, without and with E2 is shown. Results for two homozygous deletion clones are shown. C) As in (B) but assaying for *GREB1* nascent transcripts in cells deleted for the NRIP1 enhancers (left) and for (right) *NRIP1* nascent transcripts in cells deleted for the GREB1 enhancer. Two-sided Fisher exact test. n.s. not significant, *p<0.05, **p<0.01, ***p<0.001, ****p<0.0001. Statistical data are in S1 Table.
(EPS)

**S3 Fig. Related to Fig 2.** A and B) Biological replicates for the data in Fig 2A and 2B. C) Virtual 4C plots derived from the C-TALE normalized contact frequencies for the whole of the captured regions, with enhancer and promoter regions (yellow and grey bars) as viewpoints. The relative positions and size of the DNA FISH probes are shown below along with ERα +E2 ChIP-seq [16], CTCF +E2, Rad21 –E2, Rad21 +E2 ChIP-seq [49], and TT-seq +E2 30 min. Data shown for both biological replicates. Genome coordinates: hg19 human genome. D) Biological replicates for the data in Fig 2D. The statistical difference in data distribution +/-E2 at each time point was assessed by a two-sided Mann-Whitney test. n.s p >0.05, *p<0.05, **p<0.01, ***p<0.001, ****p<0.0001. Holm-Bonferroni correction for multiple testing. Statistical data are in S2 Table.
(EPS)

**S4 Fig. Related to Fig 2A**) Left; Representative images of nuclei (DAPI, blue) of MCF-7 cells untreated (-E2) or treated with E2 (+E2) for the indicated time points showing DNA FISH signal from fosmid probes targeted to the enhancer (e, green) and control (c, red) regions of the GREB1 locus. Scale bars, 5μm. Right; Violin and cumulative frequency plots showing the distribution of DNA FISH e-c fosmid interprobe distances (μm) in untreated (-E2) and E2 (+E2) treated MCF-7 cells for the indicated time points. Boxplots indicating the median distances are included. In cumulative frequency plots the red dashed lines indicate 200 nm distance, and to the right of these bar plots show the proportion of alleles with e-c inter-probe distances < or > 200nm. Data shown are for two biological replicates. B) As in (A) but for the NRIP1 locus. C) Left; Representative images of nuclei (DAPI, blue) of MCF-7 cells untreated (-E2) or treated with E2 (+E2) for 30 min showing DNA FISH signal from 10 kb probes targeted to the *GREB1* enhancer (GREB1e), and promoter (GREB1p), Scale bars, 5μm. Inset shows zoom in for probe signals. To the right; violin and cumulative frequency plots showing the distribution of DNA FISH interprobe distances (μm) between GREB1 e-p (top), and (below) control (GREB1c1 and GREB1c2) 10kb probes. The proportion (%) of e-p inter-probe distances that are < or > 200 nm are shown in the histograms to the far right. D) As in (C) but

for *NRIP1*. For the *NRIP1* promoter (NRIP1p), enhancer (NRIP1e) and intragenic (NRIP1i) 10 kb probes, and for an enhancer-control (e-c) probe pair. Statistical significance of data distributions in violin plots were assayed using a Two-sided Mann-Whitney test. n.s. p> 0.05, **p<0.01. Statistical data are in S2 Table.
(EPS)

**S5 Fig. Related to Fig 2.** A) Genome browser snapshots of the *CCND1* locus showing published ChIP-seq tracks of ERα in MCF-7 cells after E2 addition, p300 with and without E2 [16], ATAC-seq with and without E2 [23] and TT-seq tracks without E2 and 5, 30 and 60 min after E2 addition. Genome coordinates: hg19 assembly of the human genome. The position of DNA FISH probes is shown below. B) Quantification of TT-seq reads over 1kb regions extending downstream of *CCND1* and *LTO1* TSSs. Normalized counts per million reads (cpm) of two replicates are shown. p>0.05 *p<0.05, **C) Top; Representative images of nuclei (DAPI, blue) from untreated (-E2) or E2 treated MCF-7 cells showing DNA FISH signal from fosmid probes targeted to the enhancer (e, green) and promoter (p, red) regions of the *CCND1* locus. Scale bars, 5μm. Below; Violin plots showing the distribution of DNA FISH inter-probe distances (μm) between e-p fosmid probe pairs at *CCND1*cin untreated and E2 treated MCF-7 cells for the indicated time points. Boxes indicate median and interquartile distances. The statistical difference in data distribution +/-E2 at each time point was assessed by a two-sided Mann-Whitney test. n.s. p>0.05 *p<0.05, **p<0.01, ***p<0.001, ****p<0.0001. Holm-Bonferroni correction for multiple testing. Data from two independent biological replicates are shown. D) as for (C) but for enhancer-control probe pairs. Statistical data are in S2 Table.
(EPS)

**S6 Fig. Related to Fig 2.** A) C-TALE heatmaps at *GREB1* (left; 5kb resolution) and *NRIP1* (right; 10kb resolution) loci from WT and *GREB1* or *NRIP1* enhancer deletion clones without (-E2) and with E2 (+E2) for 30 min. ERα +E2 [16], TT-seq +E2 30 min, and gene tracks are shown below. In red-green-blue heatmaps, each pixel represents the normalized contact frequency between a pair of loci. In red-white-blue heatmaps each pixel represents the ratio between E2-treated enhancer deletion clones vs E2-tretead WT samples. Blue pixels indicate loss of contact frequency in enhancer deletion vs WT samples and red pixels indicate a gain. Black arrowheads indicate pixels corresponding to the enhancer-promoter pairs whose E2-dependent contact frequency is lost in enhancer deletion cells. Arrows indicate the loss of very short-range contact frequencies due to the reduction in *GREB1* transcription in enhancer deletion clones in contrast to WT cells. Genome coordinates: hg19 human genome. B) Virtual 4C plots derived from the normalized contact frequencies for replicate 2 from panel (A) using the enhancer (left) and promoter (right) regions marked with yellow and grey bars as viewpoints. C) Left; Representative images of nuclei (DAPI, blue) of WT or enhancer deletion MCF-7 clones treated as indicated showing DNA FISH signal from fosmid probes targeted to the enhancer (e, green), and the promoter (p, red) or (right) control (c, red) regions of the GREB1 or NRIP1 loci. Scale bars, 5μm. Right; Violin plots showing the distribution of DNA FISH interprobe distances (μm) between e-p probes in WT or enhancer deletion MCF-7 clones untreated or treated with E2 for 30 min. Boxplots indicating the median distances are included. Two-sided, Mann-Whitney test, Holm-Bonferroni correction for multiple testing. Biological replicate for the data in Fig 2F. D) Violin plots as for (C) but with e-c fosmid probe pairs for two replicate experiments. n.s. p>0.05, *p<0.05, **p<0.01. Statistical data are in S2 Table.
(EPS)

**S7 Fig. Related to Fig 3.** A and B) Biological replicate of C-TALE data in Fig 3D and 3E. D) Violin plots showing the distribution of DNA FISH interprobe distances of cells treated with vehicle, E2 or 4OH using e-p fosmid probes at either *GREB1* or *NRIP1* loci. Boxes indicate the median and interquartile distances. Two-sided Mann-Whitney test, Holm-Bonferroni correction for multiple testing. n.s. p>0.05, ***p<0.001, ****p<0.0001. Biological replicate for the data in Fig 3G. D) As in (C) but for enhancer (e) and control (c) probe pairs. Two biological replicates. Statistical data are in S3 Table.
(EPS)

**S8 Fig. Related to Fig 4.** A) Left: representative images of nuclei (DAPI, blue in merged) subject to immunofluorescence for ERα (green in merge) in MCF-7 cells treated with vehicle, E2 or 4OH for 30 min and pre-extracted with 0.5% Triton-X-100 before fixing. Scale bar, 10 μM. Right: boxplots showing the integrated nuclear intensity. Two-sided Mann-Whitney test, Holm-Bonferroni correction for multiple testing. B and C) Biological replicate of data in Fig 4B and 4C. D) Violin plots showing the distribution of DNA FISH enhancer-promoter interprobe distances at the *GREB1* or *NRIP1* loci in cells treated as indicated. Boxes indicating the median distances. Two-sided Mann-Whitney test, Holm-Bonferroni correction for multiple testing. Biological replicate for the data in Fig 4E. n.s p> 0.05, *p<0.05, **p<0.01, ***p<0.001, ****p<0.0001. E) Violin plots showing the distribution of DNA FISH enhancer-control interprobe at *GREB1* or *NRIP1* loci for cells treated with vehicle, FLV or TRP for 5 min prior to treatment without (-E2) and with E2 (+E2) for 30 min. Boxes indicating the median distances. Two-sided Mann-Whitney test, Holm-Bonferroni correction for multiple testing shows no significant differences. Data from two biological replicates shown. Statistical data for are in S4 Table.
(EPS)

**S1 Table. Statistical data supporting the data in Fig 1, S1 and S2 Figs, on transcriptional activation of GREB1 and NRIP1 induced by E2.**
(XLSX)

**S2 Table. Statistical data supporting the data in Fig 2, S3, S4 and S5 Figs.**
(XLSX)

**S3 Table. Statistical data supporting the data in Fig 3 and S7 Fig.**
(XLSX)

**S4 Table. Statistical data supporting the data in Fig 4 and S8 Fig.**
(XLSX)

**S5 Table. Details of gRNAs for CRISP deletions, and primers for PCR.**
(XLSX)

**S6 Table. Details of fosmid and PCR probes for DNA FISH.**
(XLSX)

**S7 Table. Details of PCR primers and BACs for C-TALE.**
(XLSX)

## Acknowledgments

We thank the Advanced Imaging Resource at the Institute of Genetics and Cancer and the Edinburgh Super-Resolution Imaging Consortium (ESRIC) for their technical support. This

work has made use of the resources provided by the Edinburgh Compute and Data Facility (ECDF) (www.ecdf.ed.ac.uk/).

## Author Contributions

**Conceptualization:** Luciana I. Gómez Acuña, Wendy A. Bickmore.

**Data curation:** Luciana I. Gómez Acuña, Wendy A. Bickmore.

**Formal analysis:** Luciana I. Gómez Acuña, Ilya Flyamer, Elias T. Friman.

**Funding acquisition:** Wendy A. Bickmore.

**Investigation:** Luciana I. Gómez Acuña, Ilya Flyamer, Shelagh Boyle.

**Methodology:** Luciana I. Gómez Acuña, Ilya Flyamer, Shelagh Boyle.

**Project administration:** Wendy A. Bickmore.

**Supervision:** Wendy A. Bickmore.

**Visualization:** Shelagh Boyle.

**Writing – original draft:** Luciana I. Gómez Acuña, Elias T. Friman, Wendy A. Bickmore.

**Writing – review & editing:** Luciana I. Gómez Acuña, Ilya Flyamer, Elias T. Friman, Wendy A. Bickmore.

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
