## [Editor Report · Decision Letter 0]

6 Dec 2023

Dear Wendy,

Thank you very much for submitting your Research Article entitled 'Transcription decouples estrogen-dependent changes in enhancer-promoter contact frequencies and spatial proximity' to PLOS Genetics.

Please move forward with a revision plan as described in your submission, and in response to the previous reviews at Review Commons. We will endeavor to submit the revised manuscript (which at PLGE will officially be an R1 even though the initial version was not evaluated at PLGE) to the same set of reviewers used for review commons.

Please aim to resubmit within the next 60 days, unless it will take extra time to address the concerns of the reviewers, in which case we would appreciate an expected resubmission date by email to plosgenetics@plos.org.

We are sorry that we cannot be more positive about your manuscript at this stage. Please do not hesitate to contact us if you have any concerns or questions.

Yours sincerely,

Gregory S. Barsh

Editor-in-Chief

PLOS Genetics

Gregory Copenhaver

Editor-in-Chief

PLOS Genetics

---

## [Decision Letter · Decision Letter 1]

29 Apr 2024

Dear Dr Bickmore,

We are pleased to inform you that your manuscript entitled "Transcription decouples estrogen-dependent changes in enhancer-promoter contact frequencies and spatial proximity" has been editorially accepted for publication in PLOS Genetics. Congratulations!

Reviewer #2 (see attached) has suggested a couple of citations that you might consider adding as you prepare your final draft for the production team - we leave that up to you, and the editorial team won't need to re-evaluate.

Yours sincerely,

Gregory P. Copenhaver

Section Editor

PLOS Genetics

Gregory Barsh

Section Editor

PLOS Genetics

Comments from the reviewers (if applicable):

Reviewer's Responses to Questions

**Comments to the Authors:**

Reviewer #1: Transcription decouples estrogen-dependent changes in enhancer- promoter contact frequencies and spatial proximity

Reviewer #2

Both reviewers shared similar concerns about the conclusions of this work. Based on the comments from both reviewers, the authors have made appropriate revisions to their manuscript and pointed out some of the caveats in their interpretation. The overall conclusion is reasonable: “Our results emphasize that the relationship between contact frequencies and physical distance in the nucleus, especially over short genomic distances, is not always a simple one”. It will be of interest to see the extent to which these observations are shared at other loci. As pointed in a recent review (Yang and Hansen, Nature reviews 2024), reconciling the relationships between chromosome conformation and imaging data is fraught with technical problems. In addition, the data presented here are not clear cut as discussed in the original reviews. It might be worthwhile to include a couple of sentences about this review and add in the reference.

Two other recent publications relevant to the topics discussed here might also be included (reviewed in Ibrahim, Nat Genetics 2024).

In conclusion, in my opinion, the work as now presented could be published to enable further discussion about the points that have been raised and to encourage others to address these important issues to establish a consensus from further observations.

**Have all data underlying the figures and results presented in the manuscript been provided?**

Reviewer #1: Yes

PLOS authors have the option to publish the peer review history of their article (what does this mean?). If published, this will include your full peer review and any attached files.

Reviewer #1: **Yes: **Douglas Higgs

**Data Deposition**

http://datadryad.org/submit?journalID=pgenetics&manu=PGENETICS-D-23-01311R1

**Press Queries**

---

## [Editor Report · Acceptance letter]

15 May 2024

PGENETICS-D-23-01311R1 

Transcription decouples estrogen-dependent changes in enhancer-promoter contact frequencies and spatial proximity 

Dear Dr Bickmore, 

We are pleased to inform you that your manuscript entitled "Transcription decouples estrogen-dependent changes in enhancer-promoter contact frequencies and spatial proximity" has been formally accepted for publication in PLOS Genetics! Your manuscript is now with our production department and you will be notified of the publication date in due course.

With kind regards,

Zsofia Freund

PLOS Genetics

On behalf of:
